# Isolated Cervical Ruptured Radiculomedullary Artery Aneurysm Predominantly Presenting as Supratentorial Subarachnoid Hemorrhage: Case Report and Review of the Literature

**DOI:** 10.3390/brainsci12050519

**Published:** 2022-04-20

**Authors:** Fuxiang Chen, Wen Lu, Baoqiang Lian, Dezhi Kang, Linsun Dai

**Affiliations:** 1Department of Neurosurgery, Neurosurgery Research Institute, The First Affiliated Hospital, Fujian Medical University, Fuzhou 350005, China; chenfuxiang@fjmu.edu.cn (F.C.); lianbaoqiang@fjmu.edu.cn (B.L.); dails@fjmu.edu.cn (L.D.); 2Department of Neurosurgery, Binhai Branch of National Regional Medical Center, The First Affiliated Hospital, Fujian Medical University, Fuzhou 350209, China; 3Fujian Provincial Institutes of Brain Disorders and Brain Sciences, First Affiliated Hospital, Fujian Medical University, Fuzhou 350005, China; 4Department of Health Management, Shengli Clinical College of Fujian Medical University, Fujian Provincial Hospital, Fuzhou 350001, China; luwen@fjmu.edu.cn

**Keywords:** spinal artery aneurysm, subarachnoid hemorrhage, vertebral artery, hybrid operating room, clipping

## Abstract

A spinal artery aneurysm originating from the branch of the extracranial vertebral artery is uncommon. Most of them were finally diagnosed due to the evidence of infratentorial or spinal subarachnoid hemorrhage (SAH). Herein, we report an extremely rare case of a ruptured spinal artery aneurysm which predominantly presented with supratentorial SAH. A 68-year-old woman was initially revealed cranial computed tomographic angiographically negative SAH with a Hunt-Hess grade of 3, while the digital subtraction angiography confirmed an isolated radiculomedullary aneurysm arising from the medial ascending branch of V2 segment at C2 level. The patient underwent surgery in a hybrid operating room. She was originally attempted with coil embolization, but successful clipping of the aneurysm was achieved through unilateral laminectomy at last. Regrettably, the current case suffered a poor clinical outcome due to the complications caused by progressive cerebral vasospasm. In summary, angiogram is of great value for this rare kind of aneurysmal definitive diagnosis. A hybrid operating room may be a feasible choice for the ruptured spinal artery aneurysm.

## 1. Introduction

Rupture of the spinal artery aneurysm is rare, especially predominantly presenting with supratentorial subarachnoid hemorrhage (SAH) [1,2]. The majority were finally diagnosed due to evidence of infratentorial or intraspinal SAH, and frequently demonstrated associations with various vascular lesions, such as arteriovenous malformation [3], dural arteriovenous fistula [4,5], moyamoya disease [6], vertebral artery (VA) occlusion [7], and coarctation of the aorta [8]. Ruptured isolated aneurysms originating from a segmental branch of the extracranial VA are extremely rare and have only been described in a few studies [9,10,11]. Currently, therapeutic strategies regarding this kind of aneurysm remain without consensus. Conservative management, surgical clipping, and endovascular embolization may be feasible and effective [1,11]. Here, we report a case of an isolated ruptured spinal artery aneurysm arising from the medial ascending branch of V2 segment, which is characterized by the presence of extensive supratentorial SAH. The patient was scheduled in a hybrid operating room (OR), where interventional treatment of the aneurysm was initially attempted, but surgical clipping was ultimately performed. In addition, a review of the pathogenesis, radiographic features, and therapeutic management are also provided.

## 2. Case Presentation

A 68-year-old woman with an unremarkable family history complained about a sudden onset of severe headache. She was admitted to the emergency department in our institution on the 1 March 2022, and a physical examination found lethargy and nuchal rigidity, suggesting a possible rupture of the intracranial aneurysm. An urgent cranial computed tomographic angiography (CTA) scan was performed, with the presence of SAH distributed mainly in the suprasellar, sylvian fissure, and prepontine cisterns, as well as in the fourth ventricle. Accordingly, the Hunt and Hess grade was 3. It also showed slight hydrocephalus with signs of ventricular dilation. However, there was no aneurysm or other vascular malformations in the intracranial arteries. Only irregular tandem stenosis or cerebral vasospasm in the intracranial segment of VA and right internal carotid artery were observed (Figure 1).

Subsequently, digital subtraction angiography (DSA) was conducted to further clarify the source of bleeding. Right VA angiography demonstrated an isolated saccular aneurysm in the medial V2 segment branch at C2 level with a diameter of approximately 3 mm. The radiculomedullary artery originated from the extracranial portion of the right VA and fed the anterior spinal artery. It clearly showed an isolated aneurysm at the junction between the radiculomedullary artery and the anterior spinal artery (Figure 2A,B). Consistent with the findings of cranial CTA, no obvious aneurysm or vascular malformations were discovered in the left VA and bilateral internal carotid arteries (Figure 2C–E).

We then scheduled this operation in a hybrid OR. Initially, we attempted to achieve endovascular coiling of the ruptured aneurysm from the right ascending cervical radiculomedullary artery instead of the straight anterior spinal artery, as we were concerned about the possibility of arterial occlusion during the intervention. A 5 French guiding catheter was first placed in the V2 segment. However, we found obvious contrast agent retention, suggesting limited blood flow caused by severe vasospasm. Then, we pulled the guiding catheter down near to the opening of the right VA. The tip of the marathon microcatheter was successfully guided into the proximal parent artery. However, it was hard to advance proximal to aneurysm because of inadequate catheter support. A microcatheter was then replaced with an Echelon-10 to provide stronger support, but it was similarly difficult to reach its target (Figure 3A). After several unsuccessful attempts, we resolutely terminated the endovascular procedure and immediately switched to operative clipping.

The patient was further placed in a side-prone position and a small incision to the back of neck was applied. The aneurysm was located at the lower border of C2 centrum (Figure 3B,C). Accordingly, a right-side laminectomy was carried out via removal of two-thirds of C2 and one-third of C3 (Figure 3D). Bloody cerebrospinal fluid flowed out quickly when the arachnoid membrane was incised under microscopic magnification. Dense intradural blood clots were found around the ruptured aneurysm during surgery and were removed. As expected, the solitary ruptured aneurysm was identified on the anterolateral side of the spinal cord. The preservation of the parent artery was quite difficult because of the small diameter, so it was clipped completely together with the aneurysm using a mini-clip (Figure 3E–G). Finally, dural closure was finished by using an artificial dura mater. Unfortunately, we did not perform an immediate angiographic review postoperatively because the arterial sheath was accidentally pulled out when deciding to change the surgical procedure. 

A scheduled CT was performed on the first postoperative day. As shown in Figure 4A–C, there was only a slight pneumocephalus, but no cerebral hemorrhage or infarction was observed, and the clip was well positioned. Furthermore, the neurological status of the patient remained basically unchanged for the first two days after the operation. However, the patient experienced a decline in consciousness at 3 days postoperatively. A head CTA confirmed aggravating cerebral vasospasm, especially bilateral VA and middle cerebral artery, and cortical hypodense changes were observed in the right cerebellum and left temporal cortex (Figure 4D–F). Therefore, we increased the strength of the anti-vasospasm treatment and performed a lumbar puncture to facilitate the drainage of bloody cerebrospinal fluid. In addition, anticoagulant therapy was also used to prevent venous thrombosis. Unexpectedly, the patient suddenly developed dyspnea, with severe oxygen desaturation and low blood pressure 7 days later. After emergency tracheal intubation and ventilator support, the patient’s vital signs gradually stabilized. However, over several hours, a sudden cardiac arrest occurred. Finally, her family refused further treatment and she died less than 1 day after leaving hospital. 

## 3. Discussion

Spinal artery aneurysm is generally involved in the anterior or posterior spinal artery, according to the previous literature [12,13,14,15], whereas spinal artery aneurysms originating from medial branch of the V2 segment are extremely rare. As shown in Table 1, there were only six cases of aneurysm arising from cervical radiculomedullary artery, including the present case, have been reported previously [9,10,11,16,17]. The mean age of these patients was 48.8 years (range: 30–68). The natural history of these aneurysms is unclear. Hemodynamic abnormalities have been considered important contributing factors to the formation of spinal artery aneurysms, owing to a large proportion of them resulting from other cerebrovascular lesions or pregnancy [6,18,19]. In addition, emerging evidence suggested that arterial wall inflammatory conditions may also be a potential pathogenesis [9,10]. For instance, cervical ruptured spinal artery aneurysm was described in a patient with Sjogren syndrome, which is an autoimmune disease which mainly leads to vasculitis [10]. Another case of spinal SAH caused by a ruptured aneurysm was the confirmed presence of mycotic features through pathological examination of the resected aneurysmal sac [9]. To date, six cases of ruptured anterior spinal aneurysms associated with VA occlusion have been reported [20], indicating that VA occlusion may be participated in the development of aneurysm. Although the VA in this patient was not completely occluded, there was severe bilateral stenosis. Therefore, we speculated that the isolated aneurysm of the present case developed due to a hemodynamic mechanism related to VA stenosis.

A retrospective review consisting of 11 patients with spinal arterial aneurysms demonstrated that sudden back pain and spinal SAH were the common clinical presentations accountable for exceed 80% [13]. Inconsistent with the present case, the patient complained of a severe headache and the appearance of extensive SAH was discovered, especially in the supratentorial cisterns. The distribution of these SAH often suggests a great possibility of internal carotid arterial aneurysm rupture. However, radiographic findings only revealed solitary aneurysm arising from the right ascending cervical radiculomedullary artery. Moreover, subarachnoid blood clots were dense around the aneurysm during operation, further supporting the fact that it was the source of rupture. The majority of spinal artery aneurysms was tiny, less than 5 mm in diameter. Therefore, CTA is probably unable to detect these lesions because of their small size and, sometimes, the scans are not low enough [21]. Currently, DSA remains the gold standard for the diagnosis of hemorrhagic cerebrovascular diseases. As described in present case, DSA has an advantage over CTA in the diagnosis of aneurysm originated from extracranial arteries supplying the spinal cord. Especially when CTA is unable to provide positive findings, the DSA examination is particularly important. 

Increasing amounts of evidence showed that endovascular intervention has become the first-line therapeutic option for intracranial aneurysms, irrespective of whether or not they rupture. Although there is no consensus of treatment with respect to spinal artery aneurysms, surgical clipping was preferentially accepted in previous studies [22,23]. This phenomenon could be related to the following reasons: Firstly, the successful embolization of an aneurysm is highly dependent on the diameter of the parent artery, which regularly is relatively small in patients with spinal artery aneurysm, resulting in a situation where it is difficult to put the microcatheter in place. Hence, the parent artery including radiculomedullary artery was obliged to be occluded to indirectly cure the aneurysm in some earlier cases [18,24]. Secondly, aneurysm embolization via the dilated anterior spinal artery approach is theoretically feasible [20], but the high risk of spinal cord ischemia may cause devastating consequences. Thirdly, a satisfactory exposition of ruptured aneurysm can be achieved easily by total or partial hemilaminectomy when the location of aneurysm was preoperatively confirmed. Despite the number of interventional embolizations for spinal cord aneurysms being not as large as that of surgical clipping, it was proved to be safe and effectively in some cases [18,20]. On the other hand, conservative treatment was also considered to be a feasible alternative strategy, since the aneurysm may disappear in a follow-up angiogram, partially due to thrombosis. Furthermore, the placement of a flow diverter in the ipsilateral VA may be another effective endovascular treatment. A previous study reported that the aneurysm disappears due to hemodynamic remodeling [25].

To our knowledge, this is the first reported case of spinal artery aneurysm treated in a hybrid OR. The advantages of a hybrid OR have been widely illustrated in previous studies [23,24], capable of performing endovascular interventions and operative surgery using anesthesia once [26,27]. As described in this case, we first implemented the coil embolization that the patient preferred, but when we found it difficult to place the microcatheter proximal to aneurysm, an alternative surgical plan was adopted. At last, the ruptured aneurysm was successfully clipped. The previous literature concerning spinal artery aneurysm suggested that most patients were able to obtain a favorable prognosis [13,20,22]. Regrettably, the current case suffered a poor clinical outcome, which may be associated with progressive cerebral or spinal vasospasm caused by a massive SAH. In fact, vasospasm management is as important as aneurysm treatment because it has a crucial effect on SAH outcome. Although the management of cerebral vasospasm, including regular monitoring of cerebral blood flow and medical treatment, had been adopted in the treatment of this patient, the effect was limited. An invasive but effective endovascular vasospasm treatment for refractory vasospasm should have been considered [28]. On the other hand, the patient had remained in bed since the SAH ictus, despite the fact that postoperative anticoagulation was used to prevent thrombosis. However, laboratory tests showed that the level of d-dimers was still significantly higher than normal. In addition, the patient mainly presented with low oxygen saturation and blood pressure when clinical deterioration suddenly happened. Taken together, pulmonary embolism could not be excluded.

## 4. Conclusions

Spinal artery aneurysm presenting with extensive supratentorial subarachnoid hemorrhage is extremely rare. DSA is of great value in diagnosis of this uncommon kind of aneurysm. In addition, a hybrid OR may be useful in their treatment.

## Figures and Tables

**Figure 1 brainsci-12-00519-f001:**
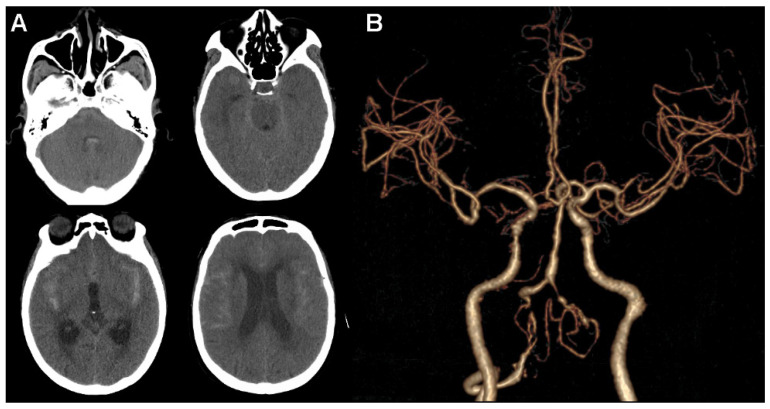
Head computed tomographic angiography scan. (**A**) Extensive subarachnoid hemorrhage is shown in axial images. (**B**) No obvious aneurysm or other vascular malformations in intracranial arteries are observed.

**Figure 2 brainsci-12-00519-f002:**
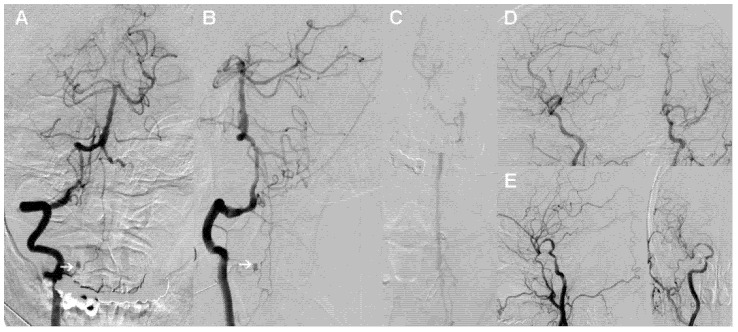
Digital subtraction angiography of the patient. (**A**,**B**) Anteroposterior (**A**) and oblique (**B**) views of the right vertebral artery show an isolated aneurysm in the medial V2 segment branch. (**C**–**E**) No obvious aneurysm or other vascular malformations in the left vertebral artery (**C**) and bilateral internal carotid arteries (**D**,**E**) observed.

**Figure 3 brainsci-12-00519-f003:**
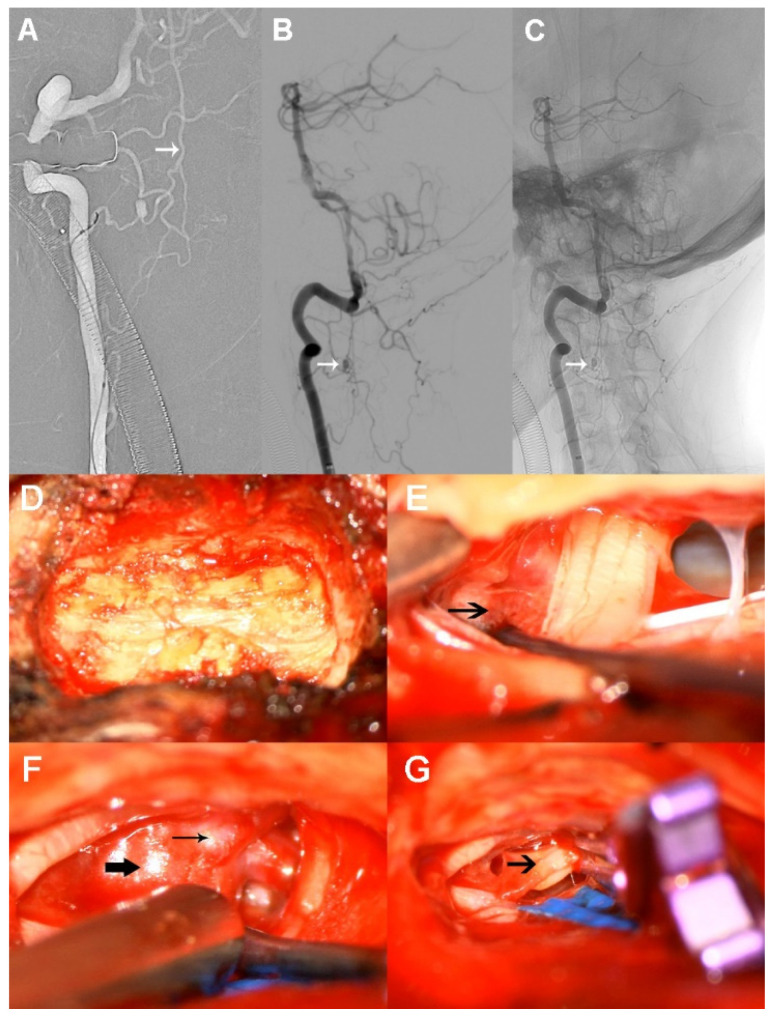
Therapy processes in the hybrid operating room. (**A**) The tip of the microcatheter is successfully guided into the proximal parent artery. (**B**,**C**) As the arrows show, the aneurysm is located at the lower border of C2 centrum. (**D**) Right-side laminectomy. (**E**) Arrow shows dense blood clots around the ruptured aneurysm. (**F**) Parent artery (thin arrow) and aneurysm (thick arrow) are shown. (**G**) The ruptured aneurysm is completely clipped, leaving the sac wall to collapse (arrow).

**Figure 4 brainsci-12-00519-f004:**
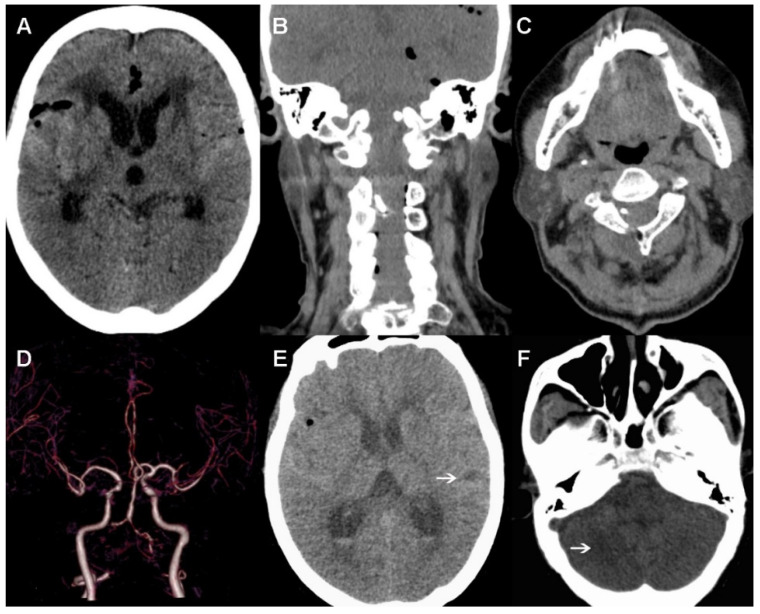
Postoperative images. (**A**–**C**) No cerebral hemorrhage or infarction is observed in the first day after surgery, and the clip is well positioned. (**D**–**F**) Cerebral vasospasm and cortical hypodense changes (indicated with arrows) found 3 days postoperative.

**Table 1 brainsci-12-00519-t001:** Clinical characteristics of aneurysm arising from cervical radiculomedullary artery.

Reference Year	Age/Sex	Symptoms	Distribution of SAH	Aneurysm		Pathogenesis	Treatment	Outcome	Angiographic Follow-Up
				Location	Type				
[16], 2000	54/man	Neck pain	Spinal	Left C4-5	Dissecting	Complication of DSA	Surgical wrapping	Good	Disappearance of aneurysm
[17], 2014	30/female	Neck pain, headache	Posterior fossa	Left C6	Saccular	Unclear	Surgical clipping	Good	Disappearance of aneurysm
[9], 2015	59/female	Headache, tetraparesis	Spinal	Left C5	Saccular	Mycotic	Surgical clipping	Death	/
[11], 2018	36/female	Back pain	Spinal	Left C6/7 and C3/4	Dissecting	Unclear	Conservative management	Good	Disappearance of aneurysm
[10], 2009	46/female	Headache	Perimedullary	Right C3	Dissecting	Sjogren syndrome -associated	Conservative management	Good	Disappearance of aneurysm
This case	68/female	Headache	Supra- and infratentorial	Right C2	Saccular	Unclear	Surgical clipping	Death	/

SAH, subarachnoid hemorrhage; DSA, digital subtraction angiography.

## Data Availability

The data presented in this report are available from the first author upon reasonable request.

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
