# Peer review of "Isolated Cervical Ruptured Radiculomedullary Artery Aneurysm Predominantly Presenting as Supratentorial Subarachnoid Hemorrhage: Case Report and Review of the Literature"

_brainsci, 2022, doi:10.3390/brainsci12050519_

Round 1

Reviewer 1 Report

Congratulation , nice, interesting report, well done a nicely documented.

Author Response

Dear reviewer,

Thank you very much for your positive comments.

Reviewer 2 Report

Dear authors,

congratulations on the nice study. The only thing requiring some improvement is the writing of the manuscript.

Present tense and simple past are intermixed, wording is sometimes unfavourable.

For example: 144/145: ..the isolated aneurysm was developed due to..

Please include the Jabbarli et al study (10.1212/WNL.0000000000007862.) as a reference for vasospasm management and point out that managing the aneurysm is as important as postinterventional intensive care medicine involving intraarterial vasospasm.

Consider discussing the use of indirect flow diversion for such aneurysms.  10.3389/fneur.2021.801470

I recall a case of an anterior spinal artery aneurysm as the cause for SAH, which was managed succesfully by implanting flow diverters in the VA across the orifice of the ASA.

Author Response

Thank you very much for your comments concerning our manuscript submitted to Brain sciences (brainsci-1667077). These comments are all valuable and very helpful for revising and improving our paper. We have tried our best to revise and improve the manuscript according to your good comments. Please see the attachment.

Reviewer 3 Report

The authors report a rare case of subarachnoid hemorrhage caused by the rupture of the radiculomedullary artery aneurysm. The theme of this report is very interesting and they can well explain and discuss the rarity and clinical importance of this case. Then, I would like to suggest some queries and propose some points.

  1. Strictly speaking, the radiculomedullary artery is not the spinal artery. Therefore, the title “Isolated cervical ruptured spinal artery aneurysm –“ might not be adequate.
  2. The CT images of the figure 1A are slight obscure for confirming the distribution of the hematoma. If possible, the authors should exchange them clearer images.
  3. I could not identify the accurate location of the aneurysm, particularly which was intradural or extradural lesion, from the presented angiograms in figure 2 and 3. Therefore, the authors should show the MPR images of the 3D angiogram of the right vertebral artery, if they performed. Otherwise, they should show the anteroposterior and lateral angiograms without subtraction for confirming the relationship between the lesion and the spinal canal.
  4. I suppose the angiogram of the figure 2B is an oblique view. If so, the authors should exchange it to the correct lateral image or change the explanation of the image about the view.
  5. Wasn’t there any possibility of the bleeding from the left vertebral artery dissection? I suggest the authors should show the angiogram of the left vertebral artery in figure 2.
  6. Line 79-83.

What size of the guiding catheter did they use? The authors assumed that the blood stagnation was caused by the vasospasm, however, it might be caused by wedge of the large caliber catheter.

  1. Figure3E-G

I could not recognize the aneurysm, the parental artery and other anatomical landmarks in these pictures. I would like the authors to show their locations in the pictures with some marks or exchange the pictures more clearer ones. Additionally, if they performed the intraoperative indochyanine green angiography with microscope, I suppose they should also show that image.

  1. If the authors performed any postoperative examinations, such as angiography, 3D-CTA and MRA, for evaluating the lesion, I suggest they should show those images.
  2. In abstract and conclusion section, the authors emphasize the usefulness of the hybrid OR in the treatment of the radiculomedullary artery aneurysm. Certainly, the intraoperative and/or postoperative angiography are useful for detecting the aneurysm location and evaluating the angiographical disappearance of the aneurysm. However, in this case, neither intraoperative and/or postoperative angiography were not performed. So, I suppose the authors should not emphasize it so much.

English usage:

Line 153, great great → great

Author Response

(The authors gave the same response as above.)

Round 2

Reviewer 3 Report

The authors responded my queries; however, I think there are still some points to improve.

  1. Generally speaking, figure legends should be described with present tense not with past tense. The authors should correct the description totally.
  2. Figure legend of figure 2.

Anteroposterior and oblique views shows – → Anteroposterior and oblique views of the right vertebral artery shows –

Which one is the angiogram of the right vertebral artery? Additionally, is it the AP image or lateral image? The authors should explain the details of each figure in the figure legends.

  1. Line 80

Guiding catheter → A 5 French guiding catheter

  1. Figure 3E

What does the arrow indicate? Explain it.

Author Response

Thank you again for your valuable comments concerning our manuscript. These comments are all valuable and very helpful for revising and improving our paper. We have tried our best to revise and improve the manuscript according to your good comments. Please see the attachment.
